

# Ferrostatin-1 inhibits fibroblast fibrosis in keloid by inhibiting ferroptosis

Liu Yang, Xiuli Li and Yanli Wang

Plastic & Cosmetics Surgery Department, Zibo Central Hospital, Zibo, China

## ABSTRACT

**Background.** Keloid is a chronic proliferative fibrotic disease caused by abnormal fibroblasts proliferation and excessive extracellular matrix (ECM) production. Numerous fibrotic disorders are significantly influenced by ferroptosis, and targeting ferroptosis can effectively mitigate fibrosis development. This study aimed to investigate the role and mechanism of ferroptosis in keloid development.

**Methods.** Keloid tissues from keloid patients and normal skin tissues from healthy controls were collected. Iron content, lipid peroxidation (LPO) level, and the mRNA and protein expression of ferroptosis-related genes including solute carrier family 7 member 11 (SLC7A11), glutathione peroxidase 4 (GPX4), transferrin receptor (TFRC), and nuclear factor erythroid 2-related factor 2 (Nrf2) were determined. Mitochondrial morphology was observed using transmission electron microscopy (TEM). Keloid fibroblasts (KFs) were isolated from keloid tissues, and treated with ferroptosis inhibitor ferrostatin-1 (fer-1) or ferroptosis activator erastin. Iron content, ferroptosis-related marker levels, LPO level, mitochondrial membrane potential, ATP content, and mitochondrial morphology in KFs were detected. Furthermore, the protein levels of α-smooth muscle actin (α-SMA), collagen I, and collagen III were measured to investigate whether ferroptosis affect fibrosis in KFs.

**Results.** We found that iron content and LPO level were substantially elevated in keloid tissues and KFs. SLC7A11, GPX4, and Nrf2 were downregulated and TFRC was upregulated in keloid tissues and KFs. Mitochondria in keloid tissues and KFs exhibited ferroptosis-related pathology. Fer-1 treatment reduced iron content, restrained ferroptosis and mitochondrial dysfunction in KFs, Moreover, ferrostatin-1 restrained the protein expression of α-SMA, collagen I, and collagen III in KFs. Whereas erastin treatment showed the opposite results.

**Conclusion.** Ferroptosis exists in keloid. Ferrostatin-1 restrained ECM deposition and fibrosis in keloid through inhibiting ferroptosis, and erastin induced ECM deposition and fibrosis through intensifying ferroptosis.

Corresponding author
Yanli Wang, wangyanliyangliu@163.com

## INTRODUCTION

Keloid is a reticular cutaneous fibroproliferative disorders (*Lee & Jang, 2018*). Keloid can occur after any dermal trauma, leading to the growth of exogenous protrusions that invade adjacent normal skin and extend beyond the original site of injury (*Limandjaja et al., 2021*). This disease is accompanied by itching, pain, joint contractures, and limited mobility,

affecting appearance and physical health, leading to psychological stress and decreased quality of life (*Lee & Seol, 2021*). Keloid occurs due to the destroyed wound healing with excessive extracellular matrix (ECM) (*Schønfeldt & Andersen, 2022*). However, the etiology of keloid is still unknown. The prevention and treatment of keloids has always been a huge challenge for burn plastic surgeons. The growing array of therapeutic options and their synergistic combinations, such as steroid therapy and laser-based devices, are yielding positive outcomes (*Ekstein et al., 2021*; *Worley et al., 2023*). Currently, the newly developing therapeutic methods such as stem cell therapy, epigenetic modifications, and gene therapy have been applied in keloids (*Naik, 2022*). Nevertheless, there is still no effective gold standard of treatment methods that yield a consistently low recurrence rate (*Ogawa, 2022*). Therefore, to find effective clinical therapies for keloid, it is urgent to explore the pathogenesis of keloid.

Studies have found that keloid formation is related to genetic, environmental, cytokine, inflammation/immunity and other factors, but its mechanism is not fully understood (*Limandjaja et al., 2020*; *Wang et al., 2020*; *Nyika, Khumalo & Bayat, 2022*; *Feng et al., 2022*). Ferroptosis is caused by the inactivation of the glutathione (GSH) dependent antioxidant system, which is manifested as iron overload, oxidative stress, excessive reactive oxygen species (ROS) generation, lipid peroxidation (LPO), and mitochondrial specific pathological manifestations (*Li et al., 2020*). Current evidence has revealed that numerous fibrotic disorders including lung, hepatic, cardiac, and renal fibrosis, is significantly influenced by cellular ferroptosis (*Zhou et al., 2022*; *Pan et al., 2021*; *Pei et al., 2022*). The excessive production of lipid peroxides induces inflammatory response and promotes fibroblasts to differentiate towards myofibroblasts, thus promoting ECM deposition and fibrogenesis (*Wang, Hua & Song, 2023*; *Liu & Wang, 2022*). It was reported that ferroptosis activator erastin promotes the differentiation of fibroblasts into myofibroblasts by inhibiting glutathione peroxidase 4 (GPX4) expression and inducing lipid peroxidation, while ferroptosis inhibitor ferrostatin-1 (fer-1) inhibits ferroptosis and fibrosis by promoting GPX4 expression and reducing lipid peroxidation (*Gong et al., 2019*). Keloid is a chronic proliferative fibrotic disease caused by a variety of factors. The main mechanism of fibrosis is the activation of fibroblasts, resulting in excessive ECM generation and deposition, which is also the basis for keloid formation (*Zhang et al., 2020*; *Barone et al., 2021*). Therefore, it is of great clinical significance to investigate ferroptosis mediated pathological mechanisms in keloid.

This study measured iron content and the expression of ferroptosis-related genes including solute carrier family 7 member 11 (SLC7A11, also known as xCT), GPX4, transferrin receptor (TFRC), and nuclear factor erythroid 2-related factor 2 (Nrf2) in keloid tissues, and explored the role of ferroptosis in the occurrence and development of keloid using a ferroptosis inhibitor ferrostatin-1 (Fer-1) and a ferroptosis activator erastin, hoping to provide experimental data for the study of the pathogenesis of scar tissue.

## MATERIALS AND METHODS

### Clinical skin tissues

The keloid tissues were collected from 70 keloid patients (26 males and 44 females, aged from 10–65 years) who were treated and operated in the orthopedics department of the Zibo Central Hospital from January 2022 to December 2022. Inclusion criteria: (a) pathological examination confirmed for keloid; (b) history of trauma, infection, surgery or scald; (c) this was the first treatment and they did not receive cryotherapy, laser treatment, hormone treatment, etc. Exclusion criteria: (a) Patients with basal cell carcinoma and other malignant tumors; (b) autoimmune diseases including pituitary or adrenal diseases, infectious diseases, skin diseases; (c) severe liver and kidney function damage; (d) skin infections and ulcers. Keloid tissues were mainly collected from earlobe, shoulder, chest, abdomen, back and perineum. In addition, 40 patients who underwent surgical treatment or skin grafting due to upper eyelid skin relaxation in the same period were selected (16 males and 24 females, aged 20–50 years old), and normal skin tissues from face and abdomen were selected as the control group. All samples obtained in this study were approved by the ethics committee of the Zibo Central Hospital (No. 202201019) and abided by the ethical guidelines of the Declaration of Helsinki. This study has received written informed consent from participants

### Primary fibroblasts isolation and culture

Primary fibroblasts isolation was conducted in reference to the reported method (*Wu et al., 2022*). The collected tissues were cut into small pieces, and digested in DMEM (Corning, NY, USA) containing 0.1 mg/mL type I collagenase (Chondrex, Woodinville, WA, USA) at 37 °C for 8 h, followed by filtration through a 200-mesh screen. The isolated fibroblasts were cultured in DMEM containing 10% fetal bovine serum (FBS; Corning, NY, USA), and 1% penicillin-streptomycin solution in 5% $CO_2$ at 37 °C. Keloid fibroblasts (KFs) and normal skin fibroblasts (NFs) from the 3–6 passage were used for subsequent experiments.

### Drug treatment

Fer-1 (Sigma, Burlington, MA, USA) and erastin (Sigma, Burlington, MA, USA) were dissolved in dimethyl sulfoxide (DMSO) and dilute with sterile distilled water. KFs were seeded into a 6-well plate ($2 \times 10^5$ cells/well) for 24 h, and then treated with the indicated final concentrations of fer-1 or erastin for 24 h. DMSO working concentration was less than 0.2%.

### Cell viability detection

KFs were seeded into 96-well plates, and then treated with a series of concentration gradients of fer-1 or erastin (0, 0.5, 1, 5, 10, 20 μM). After that, 10 μL of Cell Counting Kit-8 (CCK-8) CCK-8 reagent (Beyotime, China) were introduced. After a 2 h of incubation, the absorbance at 450 nm was tested using a microplate reader (Molecular Devices, Shanghai, China).

## RT-qPCR for mRNA expression detection

Total RNAs were extracted by using a Total RNA Kit (DP419; Tiangen, Beijing, China). Next, RNAs were reversely transcribed into cDNA using the PrimeScript RT Master Mix (TaKaRa, Dalian, China). The reaction solution was selected as 20 μL, and the temperature conditions: 37 °C, 15 min; 85 °C, 5s. Then, we used the TB Green Premix Ex Taq™ II Kit (TaKaRa, Dalian, China) to achieve PCR amplification. Two-step PCR reaction procedure was used: 95 °C, 30s, Reps: 1; 95 °C, 3s, 60 °C, 30s, Reps: 40. CT values obtained from each sample after the reaction were used to calculate the mRNA expression with the $2^{-\Delta\Delta Ct}$ method. GAPDH was used as the internal reference in this experiment. The primer sequences are shown as follows: SLC7A11 (forward, 5′- TCT CCA AAG GAG GTT ACC TGC-3′ and reverse, 5′- AGA CTC CCC TCA GTA AAG TGA C-3′); GPX4 (forward, 5′-GAG GCA AGA CCG AAG TAA ACT AC-3′ and reverse, 5′-CCG AAC TGG TTA CAC GGG AA-3′); TFRC (forward, 5′-GGC TAC TTG GGC TAT TGT AAA GG-3′ and reverse, 5′-CAG TTT CTC CGA CAA CTT TCT CT-3′); Nrf2 (forward, 5′-TCA GCG ACG AAA GAA GTA TGA-3′ and reverse, 5′-CCA CTG GTT CTG ACT GGA TGT-3′); GAPDH (forward, 5′-GGA GCG AGA TCC CTC CAA AAT-3′ and reverse, 5′- GGC TGT TGT CAT ACT TCT CAT GG-3′).

## Western blot analysis for Protein expression detection

Tissue samples (80 mg) were lysed with RIPA lysate (500 μL; Invitrogen, Carlsbad, CA, USA) and grind thoroughly for 20 min to isolate total proteins. Fibroblasts were incubated with RIPA lysate (200 μL/well) on ice for 20 min to obtain total proteins. Thereafter, proteins were separated by SDS-PAGE. The electrophoresis conditions are 80 V, 30 min, and then 120V, 60–90 min. Next, proteins were transferred to a PVDF membrane(Millipore, Bedford, MA, USA) and then blocked with 5% skimmed milk, followed by incubation with primary antibodies overnight at 4 °C and secondary antibody at 37 °C for 2 h. Then, protein bands were developed with the enhanced chemiluminescence regent (Amersham, UK), and the gray density of protein bands was analyzed by ImageJ software (National Institutes of Health, Bethesda, MD, USA). Primary antibodies used were acquired from Abcam (Cambridge, MA, USA): SLC7A11 (1:1000, ab175186), GPX4 (1:1000, ab125066), TFRC (1:1000, ab214039), Nrf2 (1:1000, ab62352), α-SMA (1:10000, ab124964), collagen I (1:1000, ab138492), collagen III (1:5000, ab7778), GAPDH (1:2500, ab9485), secondary antibody (1:2000, Abcam, ab6721).

## Iron content measurement

Proteins were extracted using the methods mentioned above. Iron content was gauged using the iron assay kit (Solarbio, Beijing, China) following the manufacturer's protocol. Briefly, cells were lysed on ice and cell supernatant was then collected by centrifugation at 10,000 g at 4 °C for 10 min. Subsequently, the iron reductase reagent and iron probe were added and incubated with the supernatant for 10 min. Intracellular ferric iron ($Fe^{3+}$) can be reduced to form $Fe^{2+}$, and $Fe^{2+}$ reacts with to iron probe produce a stable colored complex with absorbance at 593 nm. Iron content was determined by measuring the absorbance at 593 nm using a microplate reader (Bio-Rad, Hercules, CA, USA).

## LPO detection

LPO was detected as previously described. Briefly, KFs were collected and incubated with C11-BODIPY 581/591 solution (2 µM, D3861, Invitrogen, Carlsbad, CA, USA) for 30 min at 37 °C. The fluorescence emission peak shift from ~590 nm to ~510 nm after oxidation, which was detected using a flow cytometry BD LSRFortessa (Becton Dickinson, Germany). The fluorescence intensity was analyzed using FlowJo V.10.1 software.

## Transmission electron microscopy

TEM was used to observe mitochondrial ultrastructure. KFs or tissues were immersed in 2.5% glutaraldehyde at 4 °C for an entire night. Following this initial fixation, cells underwent a secondary fixation process using a 1% osmium tetroxide at room temperature for 2 h. Subsequently, the samples orderly underwent dehydration, infiltration, and embedding processes, and then were sectioned into ultrathin slices (60–80 nm). The sections were stained with 1% uranyl acetate and 0.1% lead citrate, and finally analyzed using a TEM (HT-7500; Hitachi, Japan).

## Measurement of ROS content

ROS content in KFs was measured using the Reactive Oxygen Species Assay Kit (#S0033, Beyotime, Jiangsu, China). Briefly, KFs were seeded in a 24-well plate and incubated with DCFH-DA solution at 37 °C for 20 min. Then, cells were observed under a fluorescence microscope with excitation/emission wavelengths of 488/525 nm. ROS fluorescence intensity was subjected to ImageJ software analysis.

## Measurement of MDA, GSH, and ATP levels

The treated KFs were lysed and then cell supernatant was collected *via* centrifugation at 8,000 g for 10 min at 4 °C. Malondialdehyde (MDA) level was tested with a micro MDA assay kit (Solarbio, Beijing, China) as previously described (*Zeng et al., 2024*). Briefly, the collected supernatant was mixed with the specific detection reagent and then boiled for 60 min. Then the mixture was centrifuged at 10,000 g for 10 min and added to a 96-well plate. Next, the absorbance of the samples at 532 nm were measured with a microplate reader (Bio-rad, USA). Glutathione (GSH) level was examined using a GSH assay kit (Nanjing Jiancheng Bioengineering Institute, China) as previously reported (*Jiang et al., 2023*). Briefly, standard sample and detection regents were prepared, and then mixed with the collected cell supernatant. At a mixing time of 30 s and 10 min, the absorbance of the samples at 412 nm were measured with a microplate reader. Adenosine triphosphate (ATP) contents were tested with an ATP assay kit (Beyotime, Nanjing, China). The reaction substrate and ATPase were mixed with cell supernatant at 37 °C, and the reaction was terminated after 30 min. ATP content was measured using a LKB&1250 luminometer.

## Mitochondrial membrane potential assay

A JC-1 Mitochondrial Membrane Potential Assay Kit (Solarbio, Beijing, China) was used to test the mitochondrial membrane potential of KFs referred to a reported method (*Lu et al., 2020*). Briefly, cells in each group were seeded in 24-well plates and stained with JC-1 solution for 20 min at 37 °C. Subsequently, cells were observed using a fluorescence microscope (Olympus, Tokyo, Japan).

## Immunofluorescence staining

The prepared cell slides of KFs were permeabilized for 20 min with 0.5% Triton X-100 following a 15-minute fixation in 4% paraformaldehyde. Then, cell slides were washed and blocked for 30 min using 2% goat serum. Subsequently, we added diluted primary antibody to each slide and placed it in a wet box for incubation at 4 °C overnight. Next, cell slides were incubated with fluorescent secondary antibody at room temperature for 2 h. DAPI was used as a counterstain for cell nuclei. Cell images were captured with a fluorescence microscope and analyzed by ImageJ software. The primary antibodies were shown as follows: α-SMA (1:250, ab124964, Abcam), collagen I (1:500, ab138492, Abcam), collagen III (1:500, ab7778, Abcam).

## Statistical analysis

Data from triplicate independent experiments were presented as mean ± standard deviation (SD) and analyzed using SPSS 22.0 software. All data are normally distributed and has homogeneity of variance as verified by the Shapiro–Wilk test and the Levene's test, respectively. Student's $t$-test was used for comparation of two groups, and one-way ANOVA followed by Tukey-Kramer correction was for multiple groups. $P < 0.05$ was considered statistically significant.

# RESULTS

## Iron content was elevated and ferroptosis is present in keloid tissues

To explore whether ferroptosis is involved in keloid development, we collected keloid tissues ($n = 70$) from keloid patients and normal skin tissues from healthy controls ($n = 40$), and iron content and SLC7A11, GPX4, Nrf2 and TFRC levels were determined. Our results illustrated that the iron content was substantially upregulated in keloid tissues compared with normal group (Fig. 1A) ($P < 0.001$). Then, our RT-qPCR analysis implicated that SLC7A11 ($P < 0.001$), GPX4 ($P < 0.001$), and Nrf2 mRNA ($P < 0.001$) levels were decreased and TFRC mRNA level was increased in keloid tissues (Figs. 1B–1E). Furthermore, SLC7A11 ($P < 0.001$), GPX4 ($P < 0.001$), and Nrf2 ($P < 0.001$) protein levels were significantly downregulated and TFRC protein level was ($P < 0.001$) upregulated in keloid tissues (Figs. 1F–1J). Moreover, LPO level was increased in keloid tissues compared with normal control ($P < 0.001$) (Figs. 1K–1L). TEM analysis showed that mitochondrial volume reduction, outer membrane rupture, increased membrane density, and reduced mitochondria cristae were observed in keloid tissues (Fig. 1M). Therefore, we revealed that ferroptosis is present in keloid and may be involved in in the onset and progression of keloid.

## Iron content was elevated and ferroptosis is present in keloid fibroblasts

It was observed that the iron content was substantially elevated in KFs compared with NFs (Fig. 2A). Moreover, the mRNA and protein expression of SLC7A11 ($P < 0.001$, $P < 0.001$), GPX4 ($P < 0.001$, $P < 0.001$), and Nrf2 ($P < 0.001$, $P < 0.001$) were downregulated and the mRNA and protein expression of TFRC ($P < 0.001$, $P < 0.001$) were upregulated in KFs

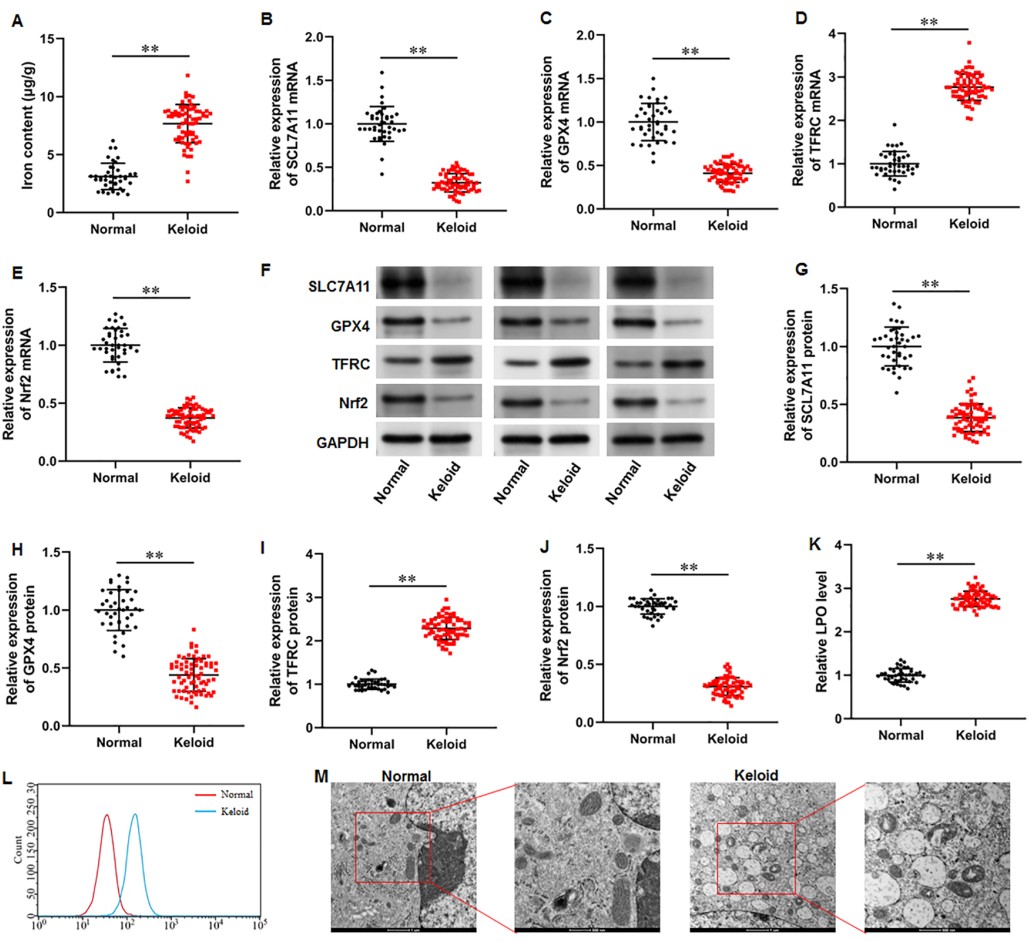

**Figure 1  Iron content was elevated and ferroptosis is present in keloid tissues.** We collected keloid tissues ($n = 70$) from keloid patients and normal skin tissues from healthy controls ($n = 40$). (A) Iron content was gauged using the iron assay kit. (B–E) SLC7A11, GPX4, TFRC, and Nrf2 mRNA levels were determined with RT-qPCR analysis. (F–J) SLC7A11, GPX4, TFRC, and Nrf2 protein levels were assessed with Western blot analysis. (K–L) LPO level was detected using C11-BODIPY staining. (M) Mitochondrial morphology was assessed using TEM. Data were presented as mean ± SD. **$P <0.01$.

compared with NFs (Figs. 2B–2J). LPO level was also increased in KFs ($P < 0.001$) (Figs. 2K–2L). Additionally, it was showed that mitochondria in KFs exhibited volume reduction, outer membrane rupture, increased membrane density, and reduced mitochondria cristae (Fig. 2M). Therefore, we revealed that ferroptosis in KFs may be associated with the pathological mechanisms of keloid.

## Ferrostatin-1 restrained ferroptosis and mitochondrial dysfunction in KFs

Fer-1 is a widely used ferroptosis inhibitor. KFs were treated with different concentrations (0, 0.5, 1, 5, 10, and 20 μM) of fer-1for 24 h. CCK-8 assay showed that fer-1 with maximum 5 μM concentration have no effect on cell viability (Fig. 3A). Thus, KFs were treated with 1 and 5 μM fer-1 for 24 h, respectively in subsequent experiments. We found that fer-1

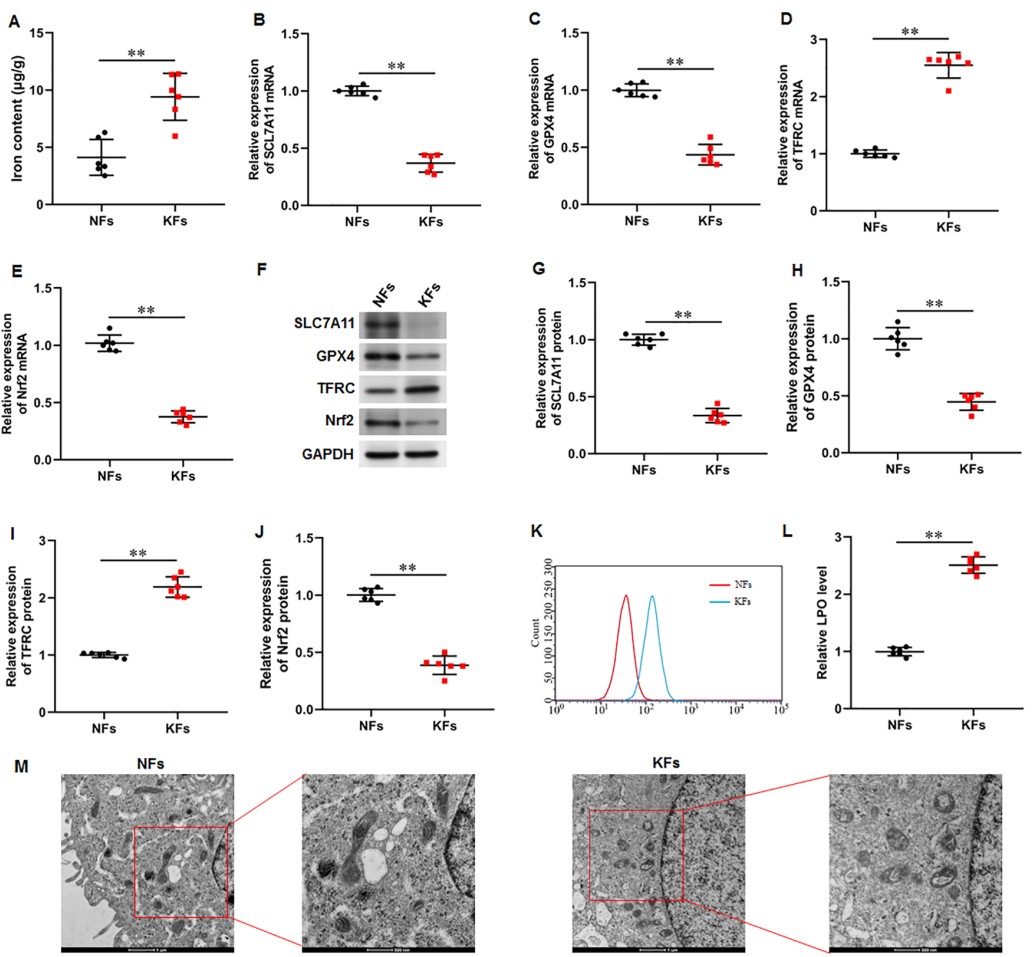

**Figure 2** **Iron content was elevated and ferroptosis is present in KFs.** (A) Iron content was gauged using the iron assay kit. (B–E) The of SLC7A11, GPX4, TFRC, and Nrf2 mRNA levels were determined with RT-qPCR analysis. (F–J) The protein levels SLC7A11, GPX4, TFRC, and Nrf2 were assessed with Western blot analysis. (K–L) LPO level was detected using C11-BODIPY staining. (M) Mitochondrial morphology was assessed using TEM. Data were presented as mean ± SD. **$P$ <0.01. (keloid fibroblasts KFs), NFs (normal skin fibroblasts).

treatment reduced iron content in KFs, and the inhibitory effect was increased with fer-1concentration ($P = 0.031$, $P < 0.001$) (Fig. 3B). Then, we found that fer-1 treatment prominently increased SLC7A11 ($P < 0.001$, $P < 0.001$), GPX4 ($P < 0.001$, $P < 0.001$), and Nrf2 protein levels ($P < 0.001$, $P < 0.001$) and decreased TFRC protein expression ($P < 0.001$, $P < 0.001$) in KFs (Figs. 3C–3G). Immunofluorescence results showed that fer-1 treatment suppressed ROS production in KFs, and 5 μM fer-1 had maximum inhibition effect ($P < 0.001$, $P < 0.001$) (Figs. 3H–3I). Moreover, fer-1 treatment decreased MDA level ($P < 0.001$, $P < 0.001$) (Fig. 3J) and LPO level ($P < 0.001$, $P < 0.001$) (Figs. 3K–3L), and increased GSH level ($P = 0.021$, $P < 0.001$) (Fig. 3M) in KFs. Additionally, JC-1 staining demonstrated that fer-1 treatment improved mitochondrial membrane potential ($P < 0.001$, $P < 0.001$) (Fig. 3N) and increased intracellular ATP content ($P < 0.001$,

$P < 0.001$) (Fig. 3O) in KFs. TEM indicated that fer-1 treatment alleviated ferroptosis-related mitochondrial pathological phenotypes in KFs (Fig. 3P). Our results revealed that fer-1 restrained ferroptosis and mitochondrial dysfunction in KFs.

### Ferrostatin-1 restrained ECM deposition and fibrosis in KFs

Numerous fibrotic disorders including lung, hepatic, cardiac, and renal fibrosis, is significantly influenced by cellular ferroptosis (*Pan et al., 2021*; *Pei et al., 2022*; *Zhang et al., 2020*). The main mechanism of fibrosis is that the activated fibroblasts leads to excessive ECM production, which is the basis for keloid formation. Thus, we investigated whether ferroptosis affect fibrosis in KFs. KFs were treated with 1 and 5 μM fer-1 for 24 h, respectively. We found that fer-1 treatment decreased α-smooth muscle actin (α-SMA) ($P < 0.001$, $P < 0.001$), collagen I ($P < 0.001$, $P < 0.001$), and collagen III protein expression ($P < 0.001$, $P < 0.001$) in KFs (Figs. 4A–4D). Likewise, our cellular immunofluorescence staining results implicated that the fluorescence intensities of these proteins were inhibited by fer-1 with all $P$ values less than 0.001 (Figs. 4E–4H). Our results revealed that ferrostatin-1 restrained ECM deposition and fibrosis through inhibiting ferroptosis in KFs.

### Erastin accelerated ferroptosis and mitochondrial dysfunction in KFs

We then used a ferroptosis activator erastin to explore ferroptosis-mediated KFs function. KFs were treated with erastin (0, 0.5, 1, 5, 10, and 20 μM) for 24 h, respectively. Our results showed that 20 μM erastin impaired the cell viability (Fig. 4A). Thus, KFs were treated with 5 and 10 μM erastin in subsequent experiments. It was implicated that erastin treatment substantially increased iron content in KFs ($P = 0.007$, $P < 0.001$) (Fig. 5B). Then, we found that erastin treatment reduced SLC7A11 ($P < 0.001$, $P < 0.001$), GPX4 ($P < 0.001$, $P < 0.001$), and Nrf2 protein levels ($P < 0.001$, $P < 0.001$) and elevated TFRC protein expression ($P < 0.001$, $P < 0.001$) in KFs (Figs. 5C–5G). Furthermore, we found that erastin treatment increased ROS fluorescence intensity ($P = 0.022$, $P < 0.001$) (Figs. 5H–5I), MDA level ($P = 0.002$, $P < 0.001$) (Fig. 5J), and LPO level ($P < 0.001$, $P < 0.001$) (Figs. 5K–5L), and decreased GSH level ($P < 0.001$, $P < 0.001$) (Fig. 5M) in KFs. Besides, JC-1 staining illustrated that erastin treatment impaired mitochondrial membrane potential ($P < 0.001$, $P <0.001$) (Fig. 5N) and decreased intracellular ATP content ($P < 0.001$, $P < 0.001$) (Fig. 5O) in KFs. Erastin treatment aggravated ferroptosis-related mitochondrial pathological phenotypes in KFs (Fig. 5P). Our results revealed that erastin accelerated ferroptosis and mitochondrial dysfunction in KFs.

### Erastin induced ECM deposition and fibrosis in KFs

We then investigated the effects of erastin-induced ferroptosis on ECM deposition and fibrosis in KFs. It was implicated that erastin treatment facilitated α-SMA ($P < 0.001$, $P < 0.001$), collagen I ($P < 0.001$, $P < 0.001$), and collagen III protein expression ($P < 0.001$, $P < 0.001$) in KFs (Figs. 6A–6D). Consistently, immunofluorescence staining results also showed the consistent results with all $P$ values less than 0.001 (Figs. 6E–6H). Therefore, erastin induced ECM deposition and fibrosis through intensifying ferroptosis in KFs.

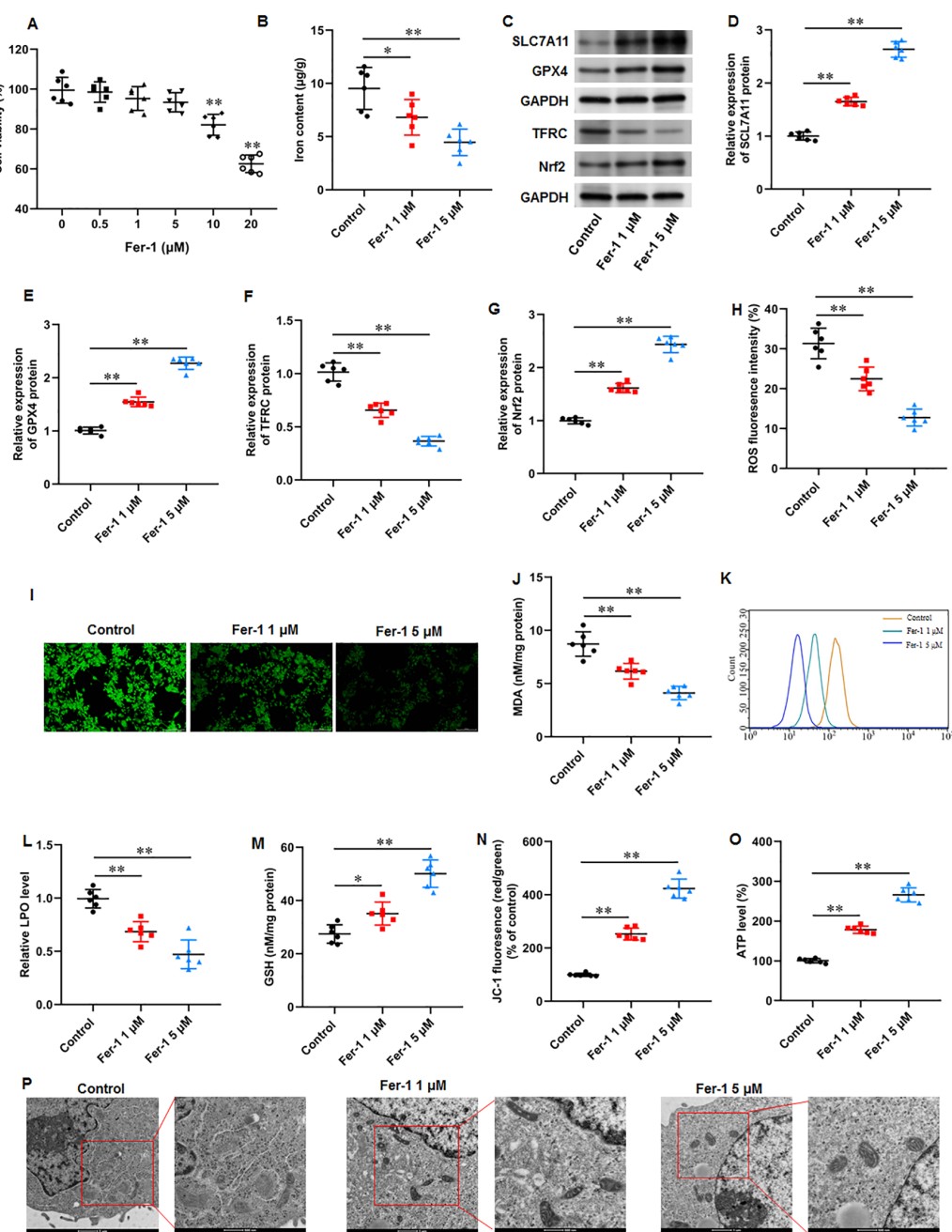

**Figure 3 Ferrostatin-1 restrained ferroptosis and mitochondrial dysfunction in KFs.** (A) The viability of KFs was assessed using CCK-8 assay after treatment with 0, 0.5, 1, 5, 10, and 20 μM fer-1. (B) Iron content in KFs was gauged using the iron assay kit. (C–G) SLC7A11, GPX4, TFRC, and Nrf2 protein levels were assessed with Western blot analysis. (H–I) ROS level in KFs was evaluated with immunofluorescence. (J) MDA level was tested with an ELISA kit. (K–L) LPO level was detected using C11-BODIPY staining. (M) GSH level was examined using an EKISA kit. (N) Mitochondrial membrane potential of KFs was determined with JC-1 staining. (O) ATP content was gauged with commercial kits. (P) Mitochondrial morphology was assessed using TEM. Data were presented as mean ± SD. *$P < 0.05$, **$P < 0.01$.

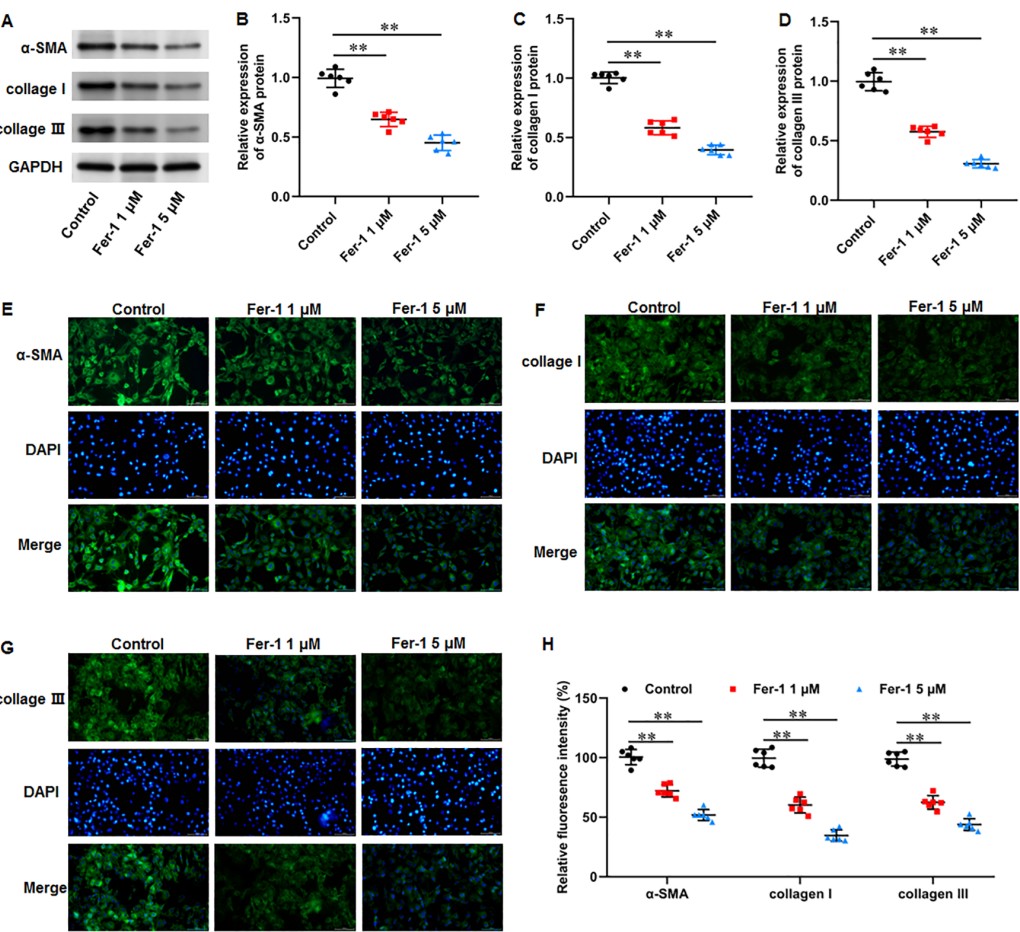

**Figure 4  Ferrostatin-1 restrained ECM deposition and fibrosis in KFs.** KFs were treated with 1 and 5 μM fer-1 for 16 h, respectively. (A–D) α-SMA, collagen I, and collagen III protein levels in KFs were tested with Western blot analysis. (E–H) Immunofluorescence staining was used to evaluate α-SMA, collagen I, and collagen III protein levels in KFs Data were presented as mean ± SD. **$P$ <0.01.

## DISCUSSION

Keloid, as a common disease in plastic surgery, is difficult to treat clinically and has a high recurrence rate. Its formation is a complex and multifactorial process. Therefore, exploring its etiology and mechanism is a contentious and challenging task in plastic surgery repair. Ferroptosis has gained significant attention in preventing and treating fibrosis. However, there are few studies on the role of ferroptosis in keloid. Therefore, it has great potential for exploration. This study aimed to explore iron content and ferroptosis-related gene expression in keloid tissues, and investigate the role of ferroptosis in the development of keloid, which will provide a new direction for exploring the pathogenesis of keloid.

Experimental researches have revealed excessive iron accumulation in fibrotic disorders. It was previously observed that patients with pulmonary fibrosis had significant amounts of iron accumulation in their lung tissues, which lead to ferroptosis, mitochondrial dysfunction, and collagen deposition (*Cheng et al., 2021*). It was found that iron overload

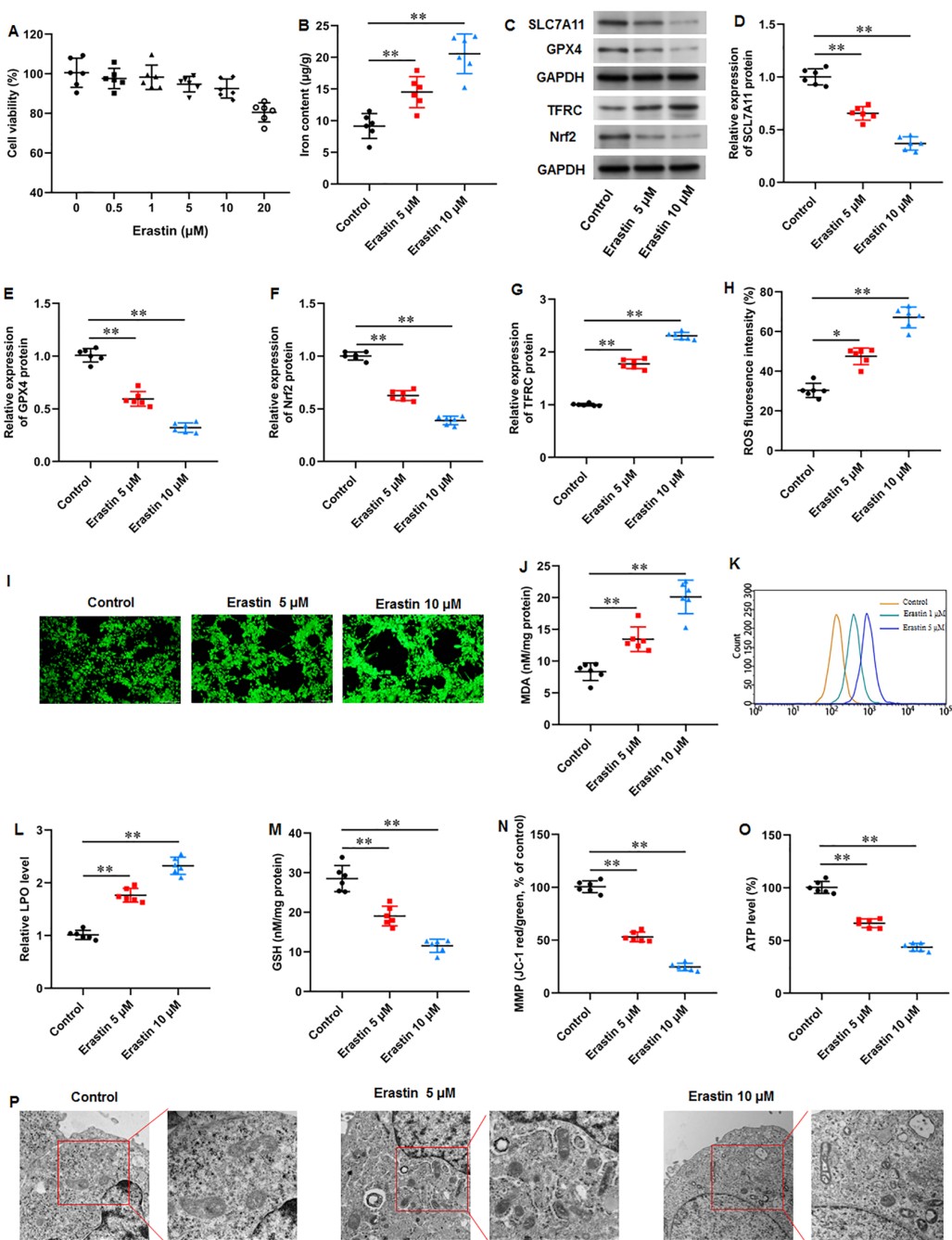

**Figure 5 Erastin accelerated ferroptosis and mitochondrial dysfunction in KFs.** (A) The viability of KFs was assessed using CCK-8 assay after treatment with 0, 0.5, 1, 5, 10 μM erastin. (B) Iron content in KFs was gauged using the iron assay kit. (C–G) SLC7A11, GPX4, TFRC, and Nrf2 protein levels were assessed with Western blot analysis. (H–I) ROS level in KFs was evaluated with immunofluorescence. (J) MDA level was examined using an ELISA kit. (K–L) LPO level was detected using C11-BODIPY staining. (M) GSH level in KFs was tested with an ELISA kit. (N) Mitochondrial membrane potential of KFs was determined with JC-1 staining. (O) ATP content was gauged with commercial kits. (P) Mitochondrial morphology was assessed using TEM. Data were presented as mean ± SD. *$P < 0.05$, **$P < 0.01$.

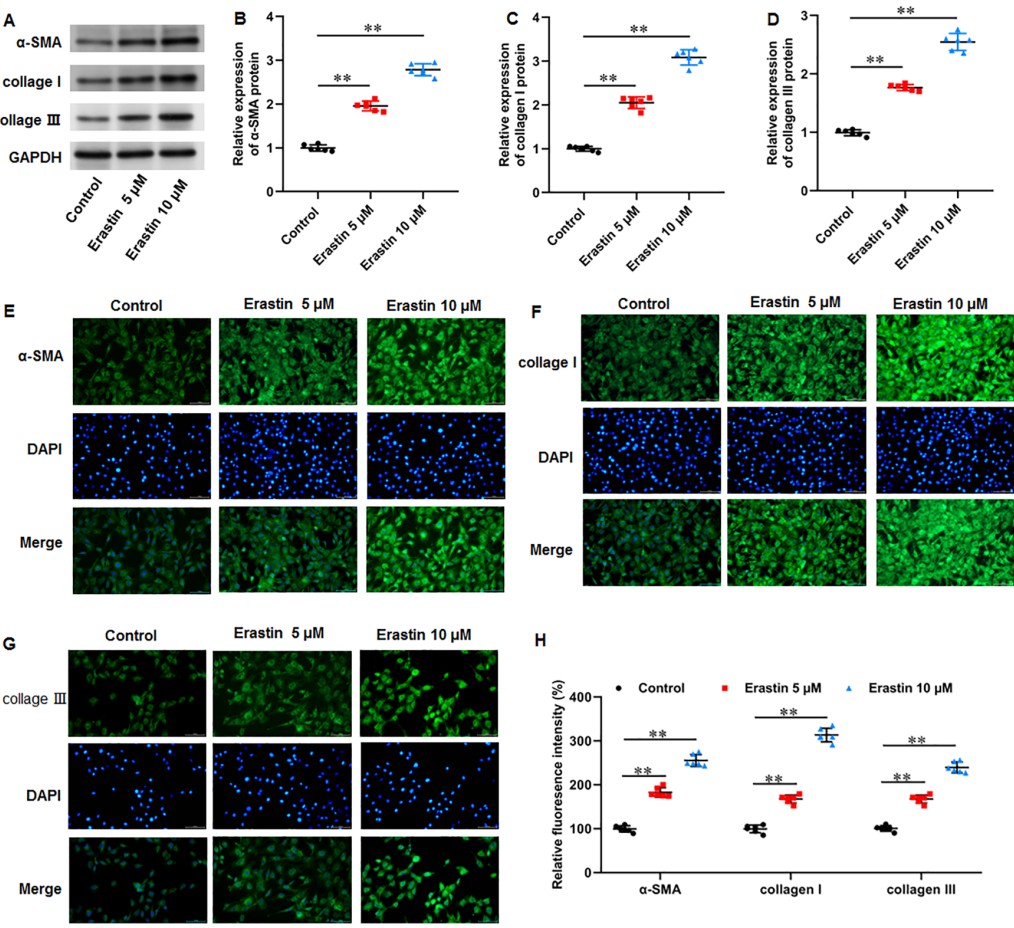

**Figure 6   Erastin induced ECM deposition and fibrosis in KFs.** KFs were treated with 5 and 10 μM erastin for 24 h, respectively. (A–D) α-SMA, collagen I, and collagen III protein levels in KFs were tested with Western blot analysis. (E-H) Immunofluorescence staining was used to evaluate α-SMA, collagen I, and collagen III protein levels in KFs Data were presented as mean ± SD. **$P <0.01$.

induced ferroptosis in hepatocytes, leading to liver injury, mitochondria damage, and liver fibrosis (*Wu et al., 2021*). Moreover, it was discovered that targeting ferroptosis is an efficient way to reduce renal fibrosis (*Liu & Wang, 2022*). These available studies suggested that ferroptosis is a momentous pathological mechanism for fibrotic diseases. Recently, several studies have mentioned the role of ferroptosis in the pathogenesis of keloid. A bioinformatics analysis has identified several novel ferroptosis-related diagnostic markers in keloid, indicating that ferroptosis is associated with the development of keloid (*Zeng et al., 2024*). Moreover, 5-aminolevulinic acid–based photodynamic therapy (5-ALA-PDT) exerted cytotoxic activity by increasing ROS and inducing ferroptosis in KFs (*Zhang et al., 2023*). Notably, iron overload is one of the characteristics of ferroptosis. Our results illustrated that the iron content was substantially elevated in keloid tissues, indicating that ferroptosis may exist in keloid.

SLC7A11 participates in extracellular cystine uptake and glutamate release, and promotes GSH synthesis. Whereas GPX4 can inactivate lipid peroxides through GSH and then inhibiting cell ferroptosis (*Liu et al., 2022*). TFRC controls the uptake of iron by cells through endocytosis, and introduces iron into the cells and stores it in ferritin. However, when a large amount of ferrous iron accumulates in cells, it induces the occurrence of cell iron death (*Kawabata, 2019*). Moreover, Nrf2 activation can reduce iron absorption, inhibit ROS production and lipid peroxide formation, thereby inhibiting ferroptosis (*Song & Long, 2020*). It has been identified that Nrf2 is a transcription factor that regulating the expression of multiple ferroptosis-related proteins. For instance, Nrf2-induced SLC7A11 transcription activation promoted radioresistance in esophageal squamous cell carcinoma through inhibiting ferroptosis (*Feng et al., 2021*). Protein arginine methyltransferase 4 (PRMT4) promoted ferroptosis by inhibiting Nrf2-mediated GPX4 transcription, thus aggravating cardiomyopathy (*Wang et al., 2022*). Moreover, ozone treatment suppressed myocardial ferroptosis by promoting Nrf2 nuclear translocation and thus initiating the expression of SLC7A11 and GPX4 (*Ding et al., 2023*). Therefore, the Nrf2/SLC7111/GPX4 pathway plays critical role in ferroptosis regulation. Our results implicated that SLC7A11, GPX4, and Nrf2 were remarkably downregulated and TFRC upregulated in keloid tissues. Therefore, we revealed that ferroptosis is present in keloid and may be involved in in the onset and progression of keloid.

It is well known that the pathological characteristics of keloid are abnormal fibroblasts proliferation and transformation to myofibroblasts expressing α-SMA, leading to excessive production of ECM rich in collagen and fibrosis development (*Tai et al., 2021*). It has been revealed that numerous fibrotic disorders are significantly influenced by cellular ferroptosis (*Pan et al., 2021*; *Pei et al., 2022*; *Zhang et al., 2020*), and targeting ferroptosis can alleviate fibrosis development. For instance, high dietary iron made mice susceptible to liver fibrosis, and fer-1 reversed this effect (*Yu et al., 2020*). Fer-1 mitigated oxalate-induced renal tubular epithelial cell fibrosis in mice by inhibiting ferroptosis (*Xie et al., 2022*). Furthermore, fer-1 treatment attenuated high fat diet-induced fibrosis, inflammation, and pathological and functional damage in liver and renal tissues (*Luo et al., 2020*). In this study, we found that fer-1 treatment decreased α-SMA, collagen I, and collagen III protein expression in KFs, while erastin treatment showed the opposite results. Our results revealed that ferrostatin-1 restrained ECM deposition and fibrosis in KFs through inhibiting ferroptosis, and erastin induced ECM deposition and fibrosis in KFs through intensifying ferroptosis.

While this research has made significant strides, it is important to acknowledge a few potential limitations. Firstly, we have the limited sample size of patients, which may affect the generalizability of our findings. Moreover, there may be have been introduced bias or confounding in the results of our experiments. In our future work, we are committed to broadening the range of our patient sample to enhance the applicability of our research. Furthermore, we will implement rigorous methodologies to identify and mitigate any potential confounding or biased elements, ensuring that our future studies provide a more comprehensive and accurate results. Additionally, we will further investigate the *in vivo*

role of ferroptosis in keloid through animal experiments, and the exploration of the related molecule mechanisms of ferroptosis in keloid is also need.

## CONCLUSIONS

Taken together, our results implicated that keloid tissues exhibited iron overload. SLC7A11, GPX4, and Nrf2 were significantly downregulated and TFRC upregulated in keloid tissues. Ferrostatin-1 restrained ECM deposition and fibrosis in KFs through inhibiting ferroptosis, while erastin induced ECM deposition and fibrosis in KFs through intensifying ferroptosis. Our findings offer significant understanding into the roles of ferroptosis in keloid. This advancement enriches our grasp of the pathophysiology of ferroptosis and fibrosis in keloid and paves the way for further investigative pursuits in this field. Our results suggest that ferroptosis inhibition in keloid represents a potentially effective clinical approach for individuals suffering from keloids, offering a new horizon for therapeutic intervention.

### Funding
The authors received no funding for this work.

### Competing Interests
The authors declare there are no competing interests.

### Author Contributions
- Liu Yang conceived and designed the experiments, performed the experiments, analyzed the data, prepared figures and/or tables, authored or reviewed drafts of the article, and approved the final draft.
- Xiuli Li conceived and designed the experiments, analyzed the data, prepared figures and/or tables, authored or reviewed drafts of the article, and approved the final draft.
- Yanli Wang conceived and designed the experiments, authored or reviewed drafts of the article, and approved the final draft.

### Human Ethics
The following information was supplied relating to ethical approvals (*i.e.*, approving body and any reference numbers):

All samples obtained in this study were approved by the ethics committee of the Zibo Central Hospital and abided by the ethical guidelines of the Declaration of Helsinki.

### Data Availability
The raw data are available in the Supplementary File.

### Supplemental Information
Supplemental information for this article can be found online at http://dx.doi.org/10.7717/peerj.17551#supplemental-information.

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
