# Peer review of "Ferrostatin-1 inhibits fibroblast fibrosis in keloid by inhibiting ferroptosis"

_PeerJ, doi:10.7717/peerj.17551_

## Round 0.1 · original submission · Major Revisions

This manuscript is of interest, and the reviewers have identified areas in need of improvement. Clearer evidence of ferroptosis role in keloid is needed eg., beyond expression changes in some proteins that have become associated with ferroptosis both in tissue and cells. Note also that there was some concern that normal fibroblast controls appear to be lacking. Please also comment on how your data sits within the current literature, including published studies mentioned by at least one of the reviewers.

**Language Note:** PeerJ staff have identified that the English language needs to be improved. When you prepare your next revision, please either (i) have a colleague who is proficient in English and familiar with the subject matter review your manuscript, or (ii) contact a professional editing service to review your manuscript. PeerJ can provide language editing services - you can contact us at [email protected] for pricing (be sure to provide your manuscript number and title). – PeerJ Staff

Reviewer 1 ·

Basic reporting

In the manuscript, Liu and colleagues tried to demonstrate that ferroptosis exists in keloid, and inhibiting ferroptosis by ferrostatin-1 could inhibit fibroblast fibrosis.

Experimental design

The topic is interesting, but the experiments need to be more well designed, and mechanism should be addressed.

Validity of the findings

1.In figure 1, I don’t think iron accumulation and changed expression of ferroptosis related proteins necessarily indicate the existence of ferroptosis in keloid.
2.In figure 2, same question exists in keloid fibrolasts.
3.In figure 3, ferroptosis restrain by Fer-1 in KFs must obtain data from LPO assay and TEM observation.
4.In figure 4-6, there are gaps between ferroptosis induction or inhibition related with ECM deposition and fibrosis. What is the mechanism?

Reviewer 2 ·

Basic reporting

Dear Author:

I would like to express my gratitude for the opportunity to serve as a reviewer for your manuscript.
Upon careful examination of your submission, I am impressed by the significance of your study. Below, I provide recommendations aimed at further strengthening your research:

Experimental design

1. RT-qPCR for mRNA Expression Detection: I recommend providing the primer sequences used for target genes to ensure transparency and reproducibility in the RT-qPCR analysis.
2. Iron Content Measurement: While the method for measuring iron content is adequately described, including the use of a commercial assay kit, providing additional information on the specific principles of the assay and any necessary sample preparation steps would enhance the method's comprehensibility and academic rigor.
3. Immunofluorescence Staining: Enhancing transparency in the Immunofluorescence staining section could be achieved by providing information on the specificity and source of primary antibodies used in the experiment, thereby strengthening the method's credibility.

Validity of the findings

4. Discussion Section: By incorporating a discussion section related to the regulation of SLC7A11 and GPX4 expression by Nrf2, the findings can be explored more comprehensively, offering new insights and directions for future research endeavors. This addition would elevate the scholarly depth of the discussion.

Additional comments

5. Although the authors have already indicated in the figure explanation that comparisons were made with the Control group, it is still advisable to explicitly mark the comparative conditions within the figure.
6. The abbreviation style for explaining KFs and NFs in the captions of Figure 2 should be standardized.
7. In Figure 3-6, please ensure that the symbols ** and ## representing the p-values are consistent. We kindly ask the authors to review carefully for any discrepancies.

Reviewer 3 ·

Basic reporting

• Overall, the paper titled "Ferrostatin-1 inhibits fibroblast fibrosis in keloid by inhibiting ferroptosis" provides a good investigation into the role of ferroptosis in keloid development. The study presents a clear background of keloid pathogenesis, emphasizing the significance of ferroptosis in fibrotic disorders.
• The overall language and formatting are appropriate, minor aspects could be addressed to improve readability and professionalism. Often authors use the adverb “obviously” when describing the results, thus implying they clearly expected to obtain that evidence. Perhaps those sentences can be rephrased.
• Literature references provided by the authors are sufficient, even if an additional relevant paper on the role of ferroptosis in keloid pathogenesis could have been mentioned (e.g. doi.org/10.1016/j.pdpdt.2023.103612). Proper background and context have been provided.

• The article is well-structured and presents findings in a logical sequence. Figures are clearly labelled, but figure legends can be improved, e.g:
- “**P<0.01, ##P<0.01, compared with control group”: here is not clear why 2 different symbols were used to indicate statistical significance).
- Densitometric quantification of protein expression by western blot experiments is not described.
• Raw data are shared. However, a clearer labelling (e.g. analysed parameters, units) would help the reader in interpreting the data.

Experimental design

• The research outlined in the paper titled "Ferrostatin-1 inhibits fibroblast fibrosis in keloid by inhibiting ferroptosis" is within Aims and Scope of the journal. A previous paper from Zhang and coll. (doi.org/10.1016/j.pdpdt.2023) explored the role of ferroptosis in keloid. However, the study from Yang et al includes a larger cohort of patients’ samples and a deeper characterisation of ferroptosis in keloid.
• Research questions are meaningful. However, while the paper discusses the potential therapeutic implications of targeting ferroptosis in keloid treatment, it could further elaborate on the clinical significance of the findings. Discussing how these results may translate into novel therapeutic strategies or clinical interventions for keloid patients would enhance the paper's impact.
• The methods outlines the collection of clinical samples, isolation of fibroblasts, and experimental procedures conducted for gene and protein expression analysis, iron content measurement, and functional assays related to ferroptosis. More details should be provided regarding the following aspects:
- Were the primary KFs and NFs generated from all the keloid tissues and all the normal skin tissues collected? Were multiple keloid tissues collected from the same patient (e.g. from earlobe, shoulder, etc.)?
- Were the tissues always utilised fresh for primary fibroblast isolation?
- A section on drug treatments (e.g.fer-1, erastin) should be added.
- In some cases the supplier is missing (e.g. RIPA buffer, enhanced chemiluminescence reagent).
- For sections 2.5, 2.6, 2.7, even if authors refer to the manufacturers’ protocols, some details about the methods should be provided, or some references using the same kits/protocols should be added (e.g. doi.org/10.3389/fnagi.2024.1309115). This would provide sufficient detail and information to replicate the results.
- The statistical analysis methods are briefly mentioned but could be elaborated upon. Providing more details on the specific statistical tests used (are data normally distributed? Are variance among different groups homogeneous?) as well as the significance levels, would strengthen the rigor of the study.

Validity of the findings

• Investigation is performed to a good technical standard. Iron content elevation in keloid tissues and fibroblasts, accompanied by dysregulation of ferroptosis-related genes, suggests the involvement of ferroptosis in keloid formation. The effects of ferrostatin-1 (fer-1) and erastin, a ferroptosis activator, on fibroblast function and ECM deposition are clearly delineated. Fer-1 treatment effectively mitigated ferroptosis, restored mitochondrial function, and reduced fibrotic markers, while erastin exacerbated ferroptosis and fibrosis.
However, one main concern is the fact that when analysing the effects of fer-1 on mitochondrial dysfunction, GSH and ROS levels, a non keloid control (e.g. normal skin fibroblast) is not included. Thus, it is difficult to understand to which extent fer-1 is restraining ferroptosis, when analysing the above parameters, in keloid samples. The same consideration can be applied to the experiments performed with erastin. Normal fibroblasts should be included in the experimental design.
• It is not clear how the concentration of fer-1 and erastin were chosen (literature, unpublished data?). Were viability assays performed to evaluate the effects of these compounds on cell viability?
• Previous scientific evidence shows that GPX4 and xCT expression levels are significantly increased in KFs compared to NFs and that treatment with 5 µM fer-1 does not affect GPX4 levels in KFs
(doi.org/10.1016/j.pdpdt.2023.103612). Can the authors try to provide a possible explanation for these different outcomes?
• In the results section, it would be useful to add p-values.
• Lines 190-191 and lines 200-201: the 2 starting sentences are exactly the same. Perhaps authors could rephrase to avoid repetition.
• Paragraphs 3.1 and 3.2 in the results section can be joined in one single section.
• In the figures, where histograms are present, dots representing the biological replicates on each bar would be useful to understand data distribution.
• Fig 1 B: asterisks overlap to SD of Nrf2 data.
• In the discussion it would be beneficial to include a section discussing the limitations of the study (e.g. in vitro research) and potential directions for future research. Addressing any constraints or unanswered questions from the current study and proposing avenues for further investigation would provide a more comprehensive outlook.

In summary, the paper presents valuable insights into the role of ferroptosis in keloid pathogenesis and highlights the potential of ferroptosis-targeted therapies in mitigating fibrosis. Addressing the suggested areas for improvement would enhance the clarity, rigor, and impact of the study.

---

## Round 0.2 · accepted · Accept

I have reviewed carefully your responses and manuscript modifications on the basis of the reviewers' feedback. I am pleased to see that these are comprehensive and to a satisfactory standard.

Reviewer 1 ·

Basic reporting

The topic is interesting, and all my questions are well addressed.

Experimental design

I have no questions.

Validity of the findings

I have no questions.